# Regulating adsorption performance of zeolites by pre-activation in electric fields

Kaifei Chen [1], Zhi Yu[1], Seyed Hesam Mousavi [1], Ranjeet Singh[1], Qinfen Gu [2], Randall Q. Snurr [3], Paul A. Webley[4] ✉ & Gang Kevin Li [1] ✉

While multiple external stimuli (e.g., temperature, light, pressure) have been reported to regulate gas adsorption, limited studies have been conducted on controlling molecular admission in nanopores through the application of electric fields (E-field). Here we show gas adsorption capacity and selectivity in zeolite molecular sieves can be regulated by an external E-field. Through E-field pre-activation during degassing, several zeolites exhibited enhanced $CO_2$ adsorption and decreased $CH_4$ and $N_2$ adsorptions, improving the $CO_2/CH_4$ and $CO_2/N_2$ separation selectivity by at least 25%. The enhanced separation performance of the zeolites pre-activated by E-field was maintained in multiple adsorption/desorption cycles. Powder X-ray diffraction analysis and ab initio computational studies revealed that the cation relocation and framework expansion induced by the E-field accounted for the changes in gas adsorption capacities. These findings demonstrate a regulation approach to sharpen the molecular sieving capability by E-fields and open new avenues for carbon capture and molecular separations.

Microporous materials generally refer to porous materials with pore diameters <20 Å. Active regulation of the pore accessibility in microporous materials is the eventual goal of advanced adsorption-based molecule processing. It has been reported that some porous materials experience framework deformation or vibration of pore-keeping groups/ions in response to temperature, light, guest accommodation, and other stimuli, thus altering the pore accessibility and improving gas separation, storage, and recognition[1].

Zeolites, composed of aluminosilicate crystalline frameworks and extra-framework cations, are widely used microporous materials in adsorption, molecular sieving, and catalytic processes. Although they are often regarded as rigid frameworks with limited flexibility, some small-pore zeolites, such as low silica potassium chabazite, K-KFI, and RHO, show a special molecular trapdoor mechanism that exclusively discriminates guest molecules[2–4]. Their pore accessibility can be regulated by the extra-framework cations which act as the "door-keeping" ions. The trapdoor cation allows the admission of gas molecules with strong polarizability or polarity, such as $CO_2$, which can sufficiently interact with the door-keeping cation to lower the energy barrier and cause the cation to deviate temporarily from the pore aperture center. It rejects "weak" gas molecules, such as $CH_4$ and $N_2$, which lack interaction ability with the cation and are unable to lower the energy barrier for pore admission[5–7]. Furthermore, a temperature increase amplifies the thermal oscillation of the trapdoor cation, increasing the admission chance of the guest molecules[8]. As a result, above a threshold temperature, the adsorption capacity of "weak" gases can anomalously increase[9].

Although the temperature is an effective stimulus to control pore accessibility, heating and cooling of the material are relatively slow due to heat transfer limitations. Inspired by the recent reports on electric-field-regulated structural transition[10–12] and molecule transport in microporous materials[13,14], we hypothesize that the application of an electric field (E-field) gradient would facilitate cation relocation and influence the access of molecules to the internal pore space of

[1]Department of Chemical Engineering, The University of Melbourne, Parkville, VIC 3010, Australia. [2]Australian Synchrotron, ANSTO, 800 Blackburn Rd, Clayton, VIC 3168, Australia. [3]Department of Chemical & Biological Engineering, Northwestern University, 2145 Sheridan Road, Evanston, IL 60208, USA. [4]Department of Chemical and Biological Engineering, Monash University, Clayton, VIC 3800, Australia. ✉e-mail: Paul.Webley@monash.edu; li.g@unimelb.edu.au

microporous materials. Ideally, the E-field required for regulating the adsorption would be below the breakdown voltage of the respective gases.

To date, understanding of E-field stimulation on gas adsorption is very limited, and experimental evidence is also rare. Although from the 1970s, it has been known that the Coulombic effect induced by external E-fields can influence the interaction between gas molecules and the surface of non-porous metal oxides[15,16], how gas adsorption in a porous medium can be affected by E-field remained unknown. Recently, some studies emerged in the area of applying E-fields to organic frameworks. A study on a membrane made of zeolitic imidazolate framework ZIF-8 indicated that a 500 V/mm E-field could reduce gas transport due to the transformation of polymorphs[13]. Some molecular simulation studies suggested certain metal organic frameworks (MOFs) with flexible frameworks and/or polar linkers, such as MIL-53 (Cr), IRMOF-1 and IRMOF-7, could undergo deformation by intense E-fields despite the concern of exceeding the breakdown voltage[10–12,17,18]. Our latest experiments demonstrated that a less intensive E-field can decrease the $CO_2$ adsorption capacity in a narrow pore MIL-53 (Al) due to the reduced charge transfer under the E-field[14].

However, to the best of our knowledge, none of the studies could produce any positive effect by using E-fields on gas sorption i.e., enhanced capacity or selectivity, which largely compromises the potential of this technique in important applications such as gas separation and molecular sieving.

In this study, we demonstrate how adsorption performance can be enhanced under stimulation by E-fields. Instead of using flexible MOFs, the subjects we chose are zeolites with rigid inorganic frameworks. We found that applying an E-field in situ during adsorption does not influence the adsorption capacity of zeolites. However, applying the E-field during pre-activation can remarkably change the subsequent adsorption capacities of $CO_2$, $CH_4$, and $N_2$, in various zeolites. Importantly, such enhancement of $CO_2$ adsorption capacity and selectivity can be maintained in multiple cycles of adsorption/desorption which is critical to pressure swing adsorption processes. We show that the relocation of the cations through an E-field polarization during pre-activation can induce lattice expansion and improve the

subsequent molecular separation ability of zeolites. The complex underlying mechanism was investigated by gas adsorption measurements, surface area analysis, synchrotron powder X-ray diffraction (PXRD) analysis, and ab initio density functional theory (DFT) calculation. To the best of our knowledge, this study is the first experimental confirmation that an external E-field during pre-activation can enhance the gas separation performance, which has significant implications for carbon capture and natural gas separations.

## Results

### E-field pre-activation of zeolites

We used two parallel electrodes to apply an external E-field perpendicular to a 0.5 mm zeolite layer (Fig. 1a), and the assembly was placed into a glass cell. The electrodes were separated by the insulated zeolite plate so that there was no current in the system and the external power source only provided a static E-field. The thin slice of the zeolite and the pair of steel plates (i.e., electrodes) allowed for fast dissipation of heat. We first tried to apply in situ E-fields during the adsorption processes of $CO_2$ and $CH_4$ but could not find any evidence that E-fields can directly control the gas uptake of zeolites (Supplementary Fig. 1) within the range of E-field gradients applied. We hypothesized that the application of the E-field during sample activation may be more successful in influencing the adsorption behavior. Accordingly, we developed a different pathway to pre-activate the sample by E-field before adsorption in which, after the zeolite sample was completely degassed to remove gas residue, an E-field was immediately applied to the highly evacuated sample at a high temperature (Fig. 1b). By this approach, a more intense E-field, up to 800 V/mm, can be applied to the zeolite, as the high vacuum condition can avoid gas breakdown caused by the large voltage gradient at low partial pressures. The latter was commonly observed for the adsorption process when the voltage was higher than 200 V. A high temperature is necessary for E-field pre-activation, as the in situ PXRD analysis indicated that the shift in XRD reflex under an E-field only occurred at high temperatures (>353 K) (see below & Supplementary Fig. 11), beyond the temperature range that is favorable for adsorption.

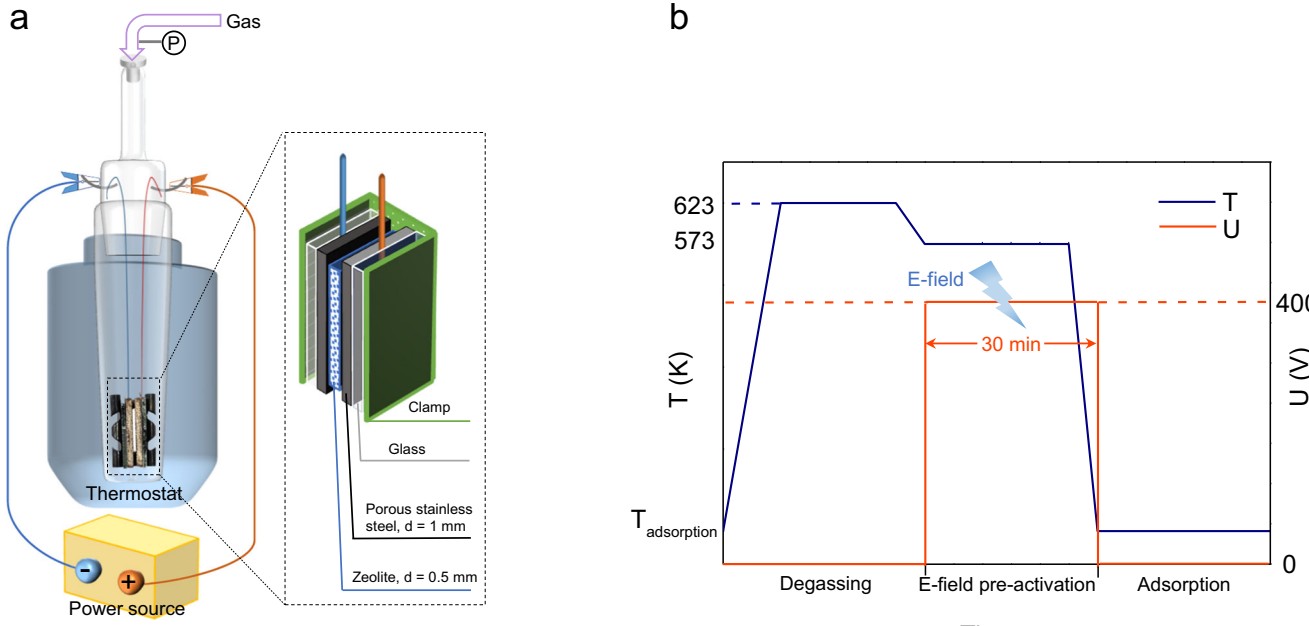

**Fig. 1 | The E-field activation of zeolite molecular sieves. a** The diagram of the experimental cell for applying the E-field on zeolites. **b** The process of E-field pre-activation and gas adsorption of potassium chabazite.

## Gas adsorption in r2KCHA regulated by E-field pre-activation

Adsorption experiments with $CO_2$, $CH_4$, and $N_2$ were performed on potassium chabazite with a Si/Al ratio of 2.2 (r2KCHA, $Si_{2.43}Al_{1.03}O_7K_{1.32}$). Before adsorption, the r2KCHA was pre-activated by an alternating E-field (4000 Hz) at the intensity of 800 V/mm at 573 K under vacuum for 30 min. All isotherms were measured at least three times with independent samples to prevent random errors from being confused with E-field effects. They had good reproducibility with random errors of <2% (as shown by the error bars in Fig. 2d).

Gas uptakes by r2KCHA without E-field pre-activation were in good accordance with literature values[3]. After the E-field pre-activation, the adsorbed amount of $CO_2$ at 273 K increased from 4.27 mmol/g to 4.42 mmol/g (Fig. 2a). In contrast, the accessibility of r2KCHA for $CH_4$ and $N_2$ was significantly reduced, with an ~20% decrease in the adsorption capacity at 100 kPa (Fig. 2b, c). The E-field effect on a specific gas is consistent at different temperatures with small errors. As shown in Fig. 2d the error bars were much smaller than the uptake difference caused by the E-field pre-activation, excluding the possibility that the effect of the E-field was raised from analysis error. Through fitting the isotherms with theoretical adsorption models (Supplementary Fig. 5), we found the maximum adsorption capacity of $CO_2$ was increased by 0.19 mmol/g by the E-field pre-activation, and those of $CH_4$ and $N_2$ decreased by 0.27 and 0.61 mmol/g, respectively. Nonetheless, the heat of adsorption for all three gases was decreased by the E-field (Supplementary Figs. 6–8). The opposite trends of gas uptake change induced by the E-field for $CO_2$ and $CH_4$ and $N_2$ provide an opportunity to improve the gas separation selectivity. The pure component selectivities[19] for component A against component B (A/B) was calculated by Eq. (1):

$$Selectivity = \frac{x_A/y_A}{x_B/y_B} \qquad (1)$$

where $x$ is the adsorbed-phase concentration and $y$ is the corresponding gas-phase concentration.

At 100 kPa and 273 K, the pure component selectivities of unactivated r2KCHA for $CO_2/CH_4$ and $CO_2/N_2$ pairs are 3.69 and 6.83, which were increased to 4.64 and 9.73, respectively, after the pre-activation of an E-field, demonstrating impressive improvements for 26% and 42%. (Fig. 2e). Notably, the selectivity enhancement at low partial pressures was even more significant (Fig. 2e).

After the thorough degassing, the E-field pre-activated r2KCHA can completely recover its adsorption capacity of $CO_2$, and partially recover the adsorption capacities of $CH_4$ and $N_2$ (Fig. 2f). It is unclear why only $CO_2$ can completely recover its adsorption capacity. We speculate that the $CO_2$ molecules have a stronger interaction with the framework and cations than the $N_2$ and $CH_4$ molecules. Therefore, $CO_2$ molecules can help to lower the energy barrier for the relaxation of any structural transition introduced by the E-field pre-activation.

In addition, a direct E-field (as opposed to an alternating field) with the same E-field intensity of 800 V/mm was applied to the r2KCHA layer to investigate if the current direction influences the effect of E-fields, as the electric charge only flows in one direction in the direct current while in the alternating current, it changes direction periodically. The uptake changes of $CO_2$ and $CH_4$ led by a direct E-field shared the same tendency as those resulting from an alternating E-field (Supplementary Fig. 9). This is to be expected as the zeolite layer was composed of isotropic powders randomly located within the layer and, on average, equally exposed to E-fields in different directions.

## Framework expansion of r2KCHA induced by E-fields

The results above raise the question of how the application of an E-field during sample pre-activation causes the change in the adsorption capacities of multiple gases. A similar E-field induced reduction in gas permeance has been reported before in the context of gas transport through a ZIF-8 membrane[13]. It was attributed to the phase transformation of the ZIF-8 lattice and the inhibition of linker rotations. The rotatable linkers in MOFs can respond to E-fields by changing the bond angles and linker orientations[17]. However, these explanations do not apply to zeolites, whose frameworks lack moveable linkages or functional groups[20]. Zeolite frameworks showed limited relaxation under pressure-induced hydration or variable temperature, and the changes in the unit cell were always accompanied by the cation relocation[21,22]. Therefore, we speculated that an E-field may impose some influence on the structure of zeolites as that observed in the above studies.

To verify this conjecture, the r2KCHA pre-activated by the E-field was analyzed by PXRD before performing adsorption measurements to avoid possible relaxation. As a comparison, the normal r2KCHA without E-field pre-activation was analyzed under the same conditions. The PXRD patterns of E-field pre-activated r2KCHA clearly showed a uniform shift of Bragg peaks to the smaller angle (Fig. 3a), suggesting an expansion of the framework[23]. This was verified by the cell volumes calculated using the cell refinement function in JADE (Fig. 3a). Synchrotron PXRD measurements with the in situ E-field application confirmed this phenomenon and further indicated that the E-field imposed a sustainable effect on r2KCHA, as removing the E-field did not immediately bring the Bragg peaks back to the original positions (Fig. 3b).

## Cation relocation in r2KCHA induced by E-fields

We seek to explain the mechanism of the E-field leading to a framework expansion and its association with the gas uptake changes. It has been widely reported that the migration of extra-framework cations can induce a relaxation of the aluminosilicate framework[21]. To date, researchers have only recognized the cation relocation in zeolites triggered by thermal oscillation[9] or guest-host interaction[3,7]. No attempt has been made to apply E-fields on the zeolite to induce cation migration. We hypothesize that the framework expansion observed here is associated with a cation relocation caused by the E-field.

To validate this hypothesis, we optimized the lattice structure of r2KCHA with the $K^+$ ions located at different sites using ab initio DFT. Chabazite is a kind of small-pore zeolite consisting of double-six ring (D6R) prisms linked by the tilted four-membered ring (4MR)[24]. The three-dimensional structure constitutes a large supercavity accessed by six eight-membered rings (8MR). Each unit cell included three D6Rs and one and a half supercavities[7]. There are four general cation positions in dehydrated chabazite[25]: site SI at the center of the D6R prism, site SII at the triad axis of the D6R prism but displaced towards the supercavity, site SIII in the supercavity above the 4MR, and site SIII′ in the 8MR (Fig. 4a). Therefore, there are three SI sites, six SII sites, nine SIII sites, and nine SIII′ sites in each unit cell. The cation number is dependent on the Si/Al ratio of zeolite, and in r2KCHA, there is a total of nine extra-framework $K^+$ in each unit cell. Since monovalent cations, such as $K^+$, energetically prefer site SIII′[25,26], the stable initial structure of r2KCHA unit cell is composed of the framework and nine $K^+$ that are all located at SIII′.

By changing the unit cell volume, we plotted the relative energy profile with one $K^+$ located at different sites through DFT calculation. When the system energy reaches the minimum, the structure of r2KCHA is the most stable. As shown in Fig. 4b, when all cations are located at SIII′, the unit cell volume of r2KCHA is 2152 Å³. A similar approach has been employed to calculate the unit cell volume of r2KCHA with one of the $K^+$ relocated from SIII′ to SI, SII, and SIII, respectively (Supplementary Fig. 12). The unit cell expands if $K^+$ relocates from the original SIII′ site to another site, which is well correlated with our observations in PXRD analysis. For instance, the unit cell volume increased to 2217 Å³ when one of the $K^+$ ions relocated to the SI site (Fig. 4c, d). The simulated PXRD patterns of r2KCHA also indicated significant left shifts that coincide with the experimental patterns (Fig. 4e). Meanwhile, the experimental PXRD patterns indicated a long-

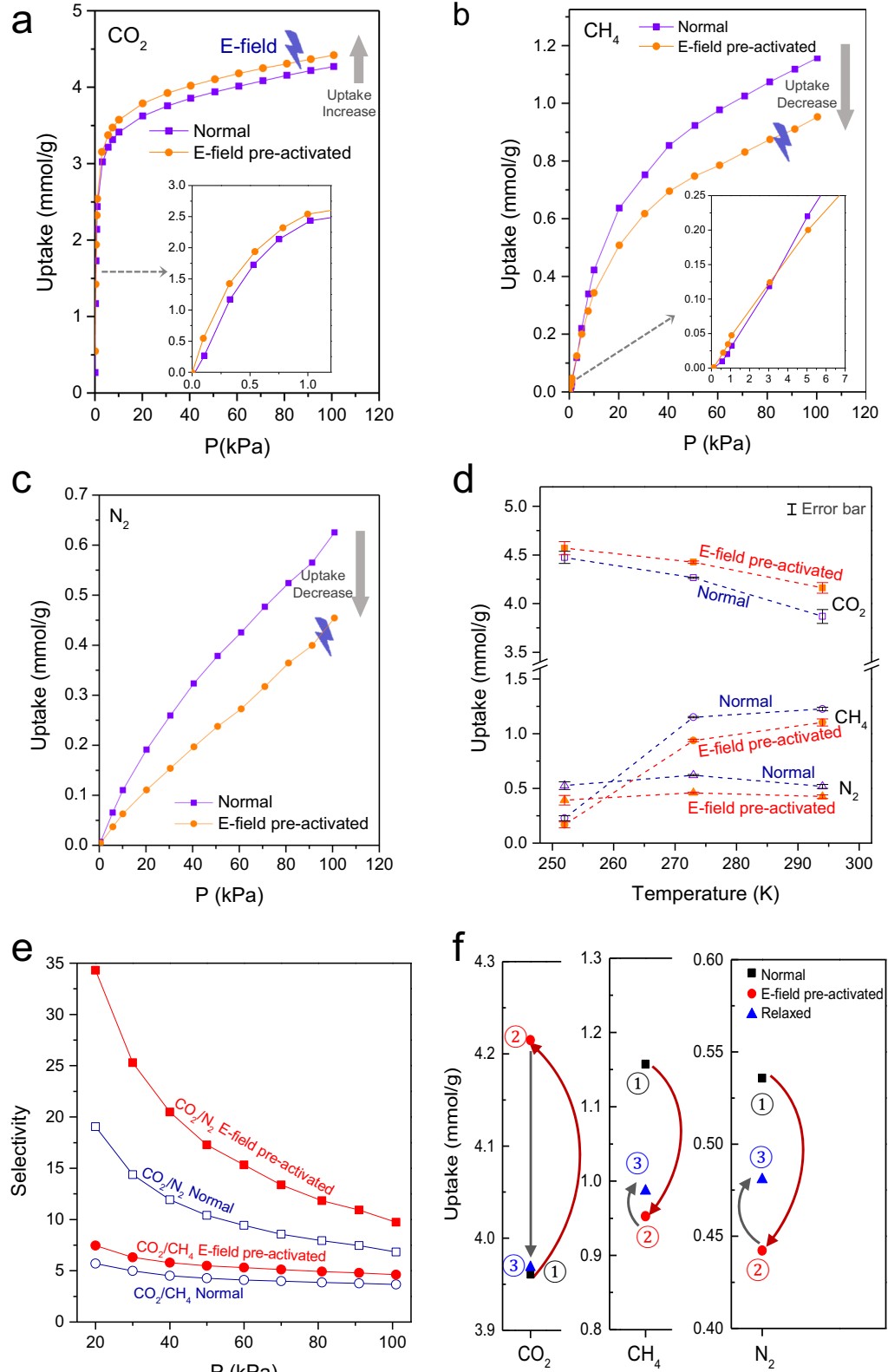

lasting peak shift (Fig. 3b), implying the cation can stay at the new site after the relocation.

By employing the Berry phase method, the spontaneous polarization of r2KCHA induced by the cation relocation was calculated. The polarization difference resulting from the relocation of one $K^+$ from SIII' to SI was 1.2, 1.1, and 2.9 $C/m^2$ along the $x$, $y$, and $z$ axes,

respectively. It was 10 times higher than the spontaneous polarization of sodalite at room temperature[27], suggesting that an E-field pre-activation can lead to a remnant polarization in r2KCHA.

Experimental evidence for the cation relocation in r2KCHA is provided by the increase of BET surface area after the E-field pre-activation. As a typical trapdoor zeolite, the $N_2$ surface area at 77 K of

**Fig. 2 | Effect of E-field pre-activation on gas adsorption in r2KCHA.** Adsorption isotherms of (**a**) $CO_2$, (**b**) $CH_4$, and (**c**) $N_2$ at 273 K without and with the E-field pre-activation. **d** Isobars of $CO_2$, $CH_4$, and $N_2$ at 100 kPa. The error bars are marked on the graph. Dash lines are for the guidance of the eye only. Complete isotherms are shown in Supplementary Figs 2–4. **e** Selectivities of $CO_2/CH_4$ and $CO_2/N_2$ were increased by the E-field pre-activation. **f** Relaxation by degassing at 623 K of the E-field pre-activated r2KCHA for $CO_2$ adsorption at 294 K and 101 kPa, $CH_4$

adsorption at 273 K and 100 kPa, and $N_2$ adsorption at 294 K and 100 kPa. ① is the first adsorption measurement without E-field pre-activation. After ①, the sample was degassed, pre-activated with the E-field, and measured for the uptake, which was labeled as ②. After ②, the sample was degassed again, and the adsorption measurement was conducted without E-field pre-activation, which was labeled as ③. The E-field intensity is 800 V/mm. Lines are guides for the eye. Source data are provided as a Source Data file.

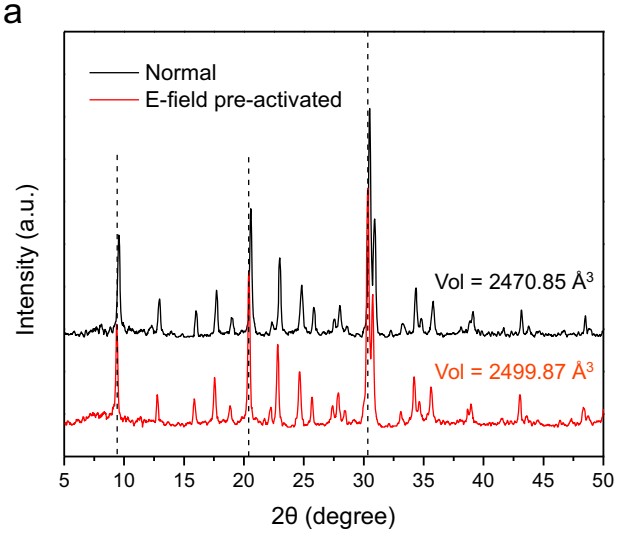

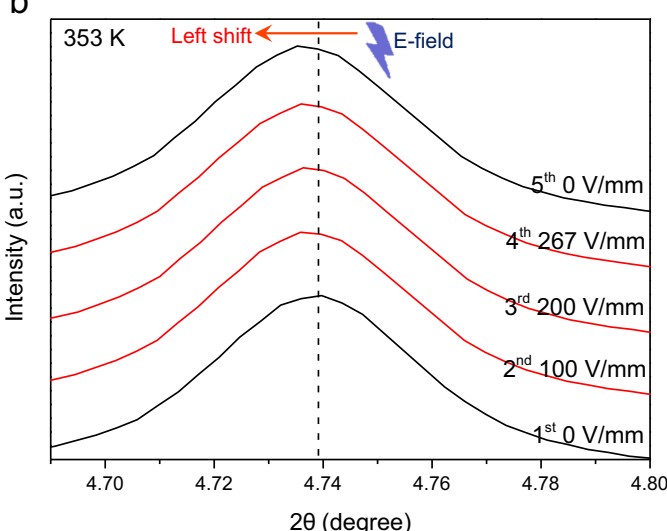

**Fig. 3 | PXRD analysis of r2KCHA. a** PXRD patterns of r2KCHA before and after the E-field pre-activation. The cell volumes were calculated by the cell refinement method of MDI Jade 6.0. The fitted profiles are shown in Supplementary Fig. 10. **b** PXRD patterns of r2KCHA with in situ E-field application at 353 K. The r2KCHA

sample was successively exposed to E-fields of 0, 100, 200, 267, and 0 V/mm. The E-field induced the left shift of the pattern. PXRD patterns at other temperatures are in Supplementary Fig. 11. Source data are provided as a Source Data file.

r2KCHA is very low, as the $K^+$ occupying the SIII' site completely blocks the passage through the 8MR at low temperatures and hinders the diffusion of $N_2$[3]. Therefore, the $N_2$ surface area of r2KCHA at 77 K is sensitive to the relocation of trapdoor cations. After being pre-activated by the E-field, the $N_2$ adsorption of r2KCHA at 77 K significantly increased (Supplementary Fig. 13) and the BET surface area doubled from 11.73 to 21.67 $m^2/g$, confirming that some of the door-keeping cations have moved away to admit the originally rejected $N_2$. According to the maximum adsorption amount of $N_2$ in r2KCHA regardless of the trapdoor effect (Supplementary Table 3), the relocated $K^+$ induced by the E-field pre-activation was estimated to be ~12%. In conclusion, it is plausible that the E-field induced framework expansion of r2KCHA was consistent with $K^+$ relocation from the original SIII' site.

**Gas admission regulated by the E-field induced cation relocation**
To explain how the cation relocation and framework expansion influence gas adsorption, we placed the gas molecules into the supercavity of r2KCHA and calculated the adsorption energies with one $K^+$ located at SIII' and SI, respectively (other $K^+$ were always at SIII'), as the $K^+$ relocation to SI could induce a moderate expansion of the framework (Fig. 4c). The adsorption energy $E_{ads}$ is defined as the difference between the total energy of the r2KCHA-gas complex and the total energy of r2KCHA augmented by the total energy of an isolated gas molecule (Eq. (2)),

$$E_{ads} = E_{tot}(r2KCHA + gas) - E(r2KCHA) - E(gas) \quad (2)$$

which represents the energy released during adsorption[28]. If we assume that one of the $K^+$ ions in the r2KCHA unit cell relocates from

the original SIII' site to the SI site after the E-field pre-activation, then the change of adsorption energy $\Delta |E_{ads}|$ caused by the E-field can be calculated by Eq. (3).

$$\Delta |E_{ads}| = |E_{ads\_SI(E-field\ pre-activated)}| - |E_{ads\_SIII'(Normal)}| \quad (3)$$

The $\Delta |E_{ads}|$ of $CH_4$ was −0.03 eV, suggesting the energy released during $CH_4$ adsorption will be lower if the $K^+$ relocates from SIII' to SI, which is consistent with the decreased heat of adsorption obtained from experimental isotherms (Supplementary Figs. 6, 8 and Supplementary Table 2). The decrease in adsorption energy will reduce the gas adsorption capacity, which explains the situation observed in $CH_4$ and $N_2$ adsorption. As shown in Fig. 5a, b, the charge rearrangement (the enrichment or depletion of the electron density) after $CH_4$ adsorption appears to be weakened when the $K^+$ relocates from SIII' (Fig. 5a) to SI (Fig. 5b). It suggested that compared with the $K^+$ at SIII', the $CH_4$ molecule had a more-limited interaction with the $K^+$ located at SI, which contributed to the decreased adsorption energy after the $K^+$ relocation. Meanwhile, the framework expansion induced by the $K^+$ relocation (Fig. 3a) will also reduce the affinity between the host framework and the guest molecules, as an expanded framework will decrease the electron density around the adsorption sites[29].

As "weak" gases with little interaction with cations, $CH_4$ and $N_2$ are excluded by pore-blocking $K^+$ when the temperatures are lower than the critical admission temperature ($T_c$)[8]. Theoretically, the relocation of trapdoor $K^+$ can lead to the successful passage of gas molecules through the 8MR, facilitating the adsorption of $CH_4$ and $N_2$. However, in our cases, such a gate-opening effect caused by the E-field induced cation relocation was not observed, as the adsorption capacities of $CH_4$ and $N_2$ after E-field pre-activation were decreased. It can be

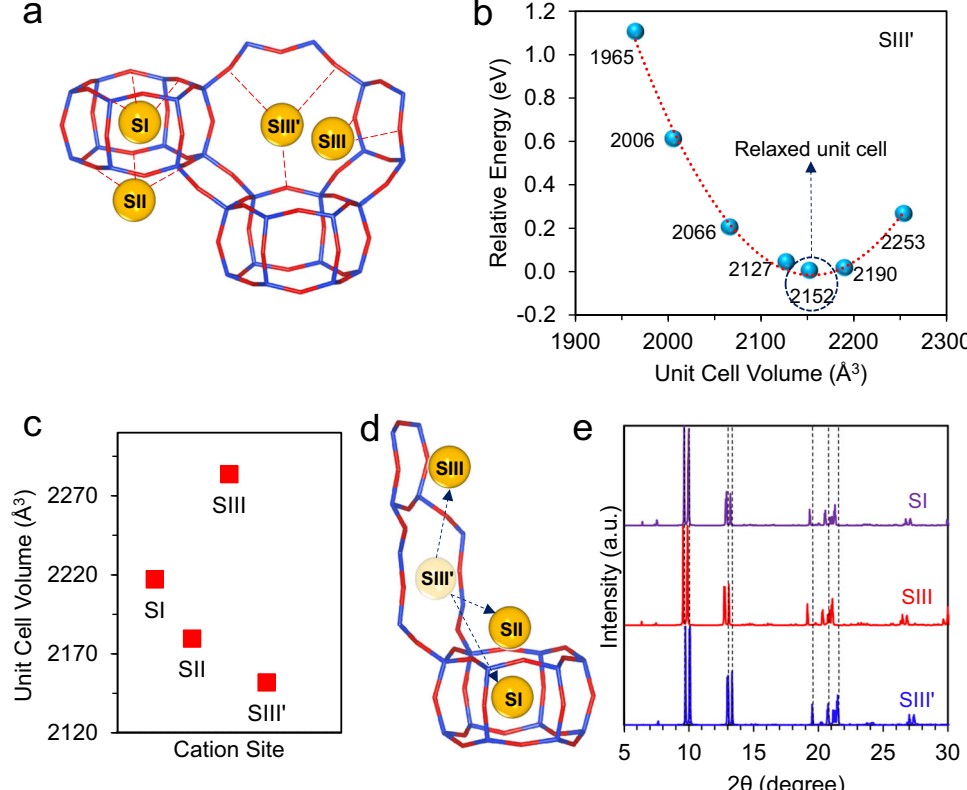

**Fig. 4 | Framework expansion of r2KCHA induced by cation relocation. a** The schematic illustration of chabazite with four cation sites. Yellow: Potassium. Blue: Silicon or Aluminum. Red: Oxygen. **b** Changes of the relative energy over unit cell volume with all $K^+$ located at SIII'. **c** The unit cell volumes of r2KCHA when one of the $K^+$ locates at different cation sites. **d** The pathways for the $K^+$ relocating from SIII' to other cation sites. Yellow: Potassium. Blue: Silicon or Aluminum. Red: Oxygen. **e** The simulated PXRD patterns of r2KCHA unit cell with eight $K^+$ ions located at SIII' site and one $K^+$ ion located at SI, SIII and SIII' site, respectively. Source data are provided as a Source Data file.

explained for two reasons. Firstly, the relocation of partial $K^+$ cannot fully allow gas molecules to diffuse into the internal cavities, because other $K^+$ that stay at SIII' will block the diffusion pathway. Although the BET surface area of r2KCHA increased after the E-field pre-activation, it was still far less than normal microporous materials, implying not all of the trapdoor $K^+$ were relocated. Secondly, the adsorption temperatures (252, 273, and 294 K) were beyond the $T_c$ of $CH_4$ (250–279 K) and $N_2$ (243–266 K)[8], at which the pore-blocking $K^+$ has already temporarily deviated by thermal oscillation[9]. Therefore, the influence of the expanded framework and the decreased adsorption energy will dominate to decrease the adsorption capacity. Once the adsorption temperature is low enough, the gate-opening effect led by the cation relocation will be dominant to increase the adsorption capacity, as in the case of $N_2$ adsorption in r2KCHA at 77 K (the BET surface area analysis).

As for $CO_2$, the $|E_{ads}|$ from DFT was decreased by 0.013 eV after the E-field pre-activation, which was in agreement with the decrease in the heat of adsorption calculated from experimental isotherms (Supplementary Figs. 5, 8 and Supplementary Table 1). However, it seemed contradictory to the increased adsorption capacity of $CO_2$ after the E-field pre-activation. Given this difference, an increase in adsorption sites due to the relocation of cations was considered to account for the improvement of $CO_2$ adsorption capacity. It has been reported that $CO_2$ molecules will prefer to occupy the center of 8MR in zeolites while $N_2$ cannot occupy that site[30]. When an E-field was applied, some of the $K^+$ located at SIII' moved away, leaving a space for the $CO_2$ molecule. We studied the system energies when the $CO_2$ molecule passes through the unblocked 8MR and diffuses into the supercavity. When the $CO_2$ occupies the 8MR, the system energy is the lowest (Fig. 5c),

demonstrating that the SIII' originally occupied by the $K^+$ is a new adsorption site of $CO_2$. In comparison, the $CH_4$ molecule cannot stay at the center of the 8MR, and its adsorption site remains inside the supercavity even if the SIII' site is available (Fig. 5c). The reason is that the diameter of the 8MR in r2KCHA was ~3.8 Å, which is too small for $CH_4$ (kinetic diameter = 3.8 Å) to locate, while $CO_2$ (kinetic diameter = 3.3 Å) can more freely stay at the window site. Meanwhile, $CO_2$ molecules have a stronger quadruple interaction with the framework than $CH_4$ molecules. Both factors made the center of the 8MR only ideal for $CO_2$[30,31].

### Demonstration for other zeolites

In addition to r2KCHA, some other zeolites were investigated, including trapdoor zeolites ZSM-25-Na ($Na_{330}[H_{30.7}Al_{360.7}Si_{1079.3}O_{2880}]$)[32], ZSM-25-K ($K_{327.9}Na_{2.1}[H_{30.7}Al_{360.7}Si_{1079.3}O_{2880}]$)[33], and the ionic liquid zeolite TMA-Y[34,35] without trapdoor effect. The PXRD patterns of ZSM-25-K and TMA-Y under an E-field showed analogous shifts to the smaller angle as that of r2KCHA (Supplementary Figs. 14 and 15), suggesting that the E-field pre-activation can impose similar structural variations on different zeolites. A similar change in adsorption capacity induced by an E-field was also observed. Overall, the $CO_2$ adsorption capacity of zeolites can be enhanced by E-fields while the adsorption of $CH_4$ is commonly suppressed as shown in Fig. 6a, b. This general pattern is suggestive of an attainable approach to improving molecular separation selectivity by E-fields.

The exception was the $CH_4$ uptake in ZSM-25-K at 252 K and 273 K, which significantly increased after E-field pre-activation (Fig. 6c and Supplementary Fig. 18). At 252 K and 273 K, which are below the $T_{c(CH4)}$ of ZSM-25-K (275–350 K), the $K^+$ completely

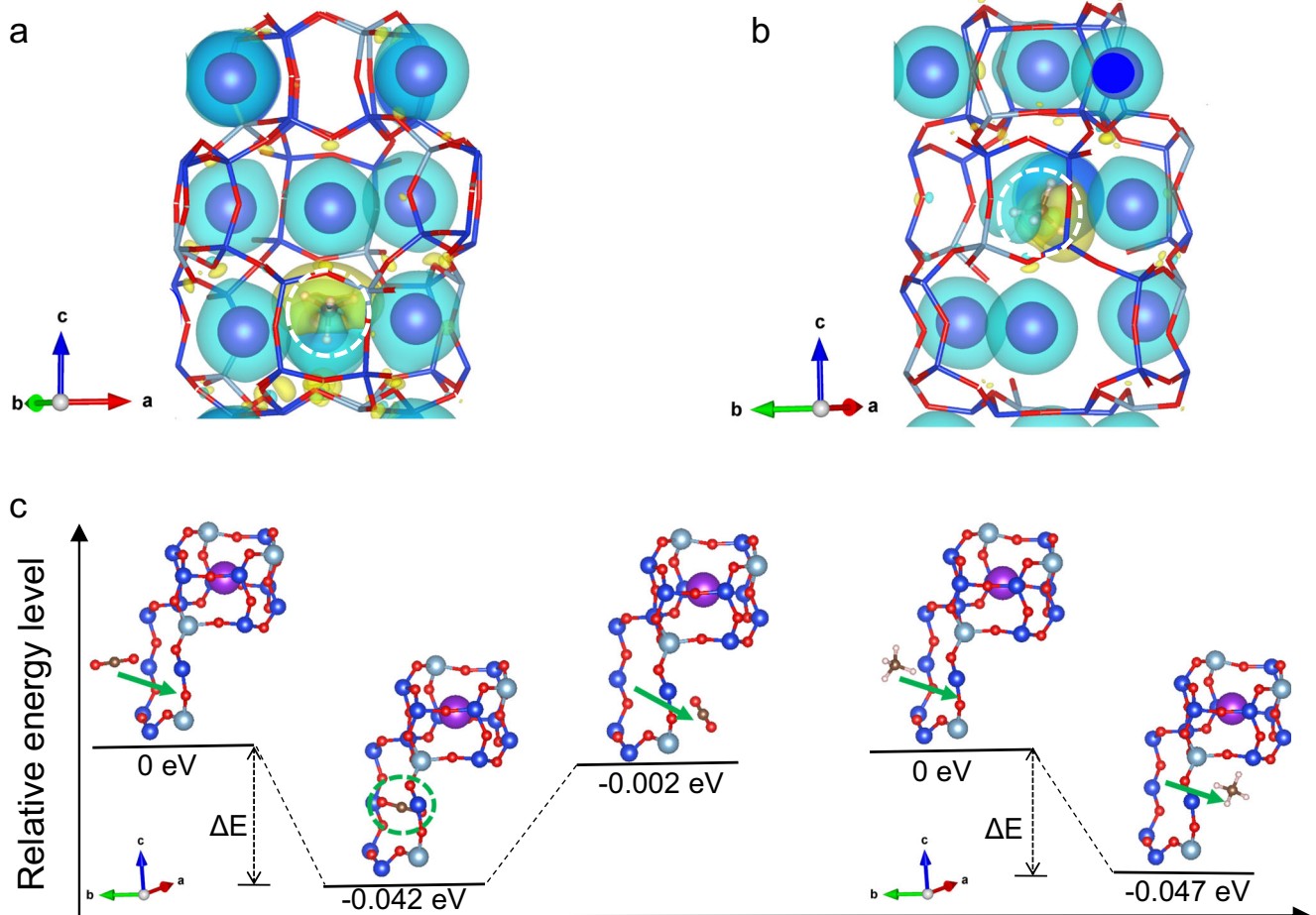

**Fig. 5 | Gas adsorption in r2KCHA simulated by the ab initio DFT calculation. a, b** Electron density difference of the $CH_4$-CHA complex with (**a**) all $K^+$ located in SIII' or (**b**) one of the $K^+$ relocated from SIII' to SI. The yellow iso-surfaces represent the electron accumulation region, and the blue iso-surfaces represent the electron depletion region. The iso-surfaces are plotted at 0.0001 $e^-$/bohr³. **c** The relative energy level of r2KCHA with one $CO_2$ (left) or $CH_4$ (right) molecule passing through the unblocked 8MR. In r2KCHA, one of the $K^+$ relocates from SIII' to SI, leaving a space at the center of the 8MR. Purple: Potassium. Blue: Silicon. Silver: Aluminum. Red: Oxygen. Brown: Carbon. White: Hydrogen.

blocked the 8MR of ZSM-25, leading to the low adsorption capacity of $CH_4$[33]. Therefore, the relocation of trapdoor $K^+$ induced by the E-field pre-activation can substantially reduce the energy barrier of $CH_4$ molecules to pass through the 8MR, which dominated over the decreased adsorption energy caused by the framework expansion and finally increased the $CH_4$ uptake. This phenomenon was opposite to that observed in r2KCHA, in which the $CH_4$ adsorption was reduced by the E-field pre-activation at the same temperatures. It is due to the particularity of trapdoor zeolites as the temperature is a crucial factor for oscillation and relocation of trapdoor cations[9]. The original state of trapdoor cations before E-field pre-activation will decide the priority of two opposite consequences brought by the cation relocation: the gate-opening effect that facilitates adsorption, and the decreased adsorption energy that impairs adsorption. For r2KCHA, the $T_{c(CH4)}$ was 250–299 K[8], implying that the trapdoor $K^+$ was partially open above 250 K, which diminished the gate-opening effect and emphasized the influence of the expanded framework after the E-field pre-activation.

Based on the experimental phenomena and computational simulation, we believe that an E-field can facilitate the cation relocation in zeolites. As demonstrated in previous studies, in the presence of an external stimulus, such as a high temperature, the cations will be able to overcome the energy barrier to relocate to a local minimum[9]. The E-field serves as the stimulus to energize the cation relocation in a way similar to a temperature-induced stimulus. When the cation relocates

from its pristine location under an E-field, it can hold at its new location after the E-field was removed, because there will be an energy barrier for the cation to move back, as shown by the calculated energy profile in Supplementary Fig. 21. The E-field effect was reversible after the high-temperature degassing (Fig. 2f). However, degassing the zeolite at a low temperature may prolong the effect of E-field activation as the moderate temperature cannot facilitate the relocated cations to overcome the energy barrier to relax to its pristine location. This assumption was validated on TMA-Y. By degassing the E-field pre-activated TMA-Y at 393 K, the E-field effect on TMA-Y can be retained for at least 5 cycles of adsorption-desorption, creating a high and long-lasting selectivity of $CO_2$/$CH_4$ (Fig. 6d). The decreased $CO_2$ uptake in the third batch of adsorption resulted from the lower degassing temperature, which was the case in the TMA-Y without E-field pre-activation as well. The lasting effect makes the E-field pre-activation applicable in the industrial adsorption process, such as pressure swing adsorption.

## The effect of E-field pre-activation on gas separation

Column breakthrough experiments of TMA-Y for the mixture of $CO_2$/$CH_4$ (50:50, v/v) at 298 K at atmospheric pressure were conducted to evaluate the effect of E-field pre-activation on gas separation. During the breakthrough process, $CH_4$ eluted at the start of the experiment while $CO_2$ was detected later at the outlet gas. The E-field pre-activation of TMA-Y impaired the $CH_4$

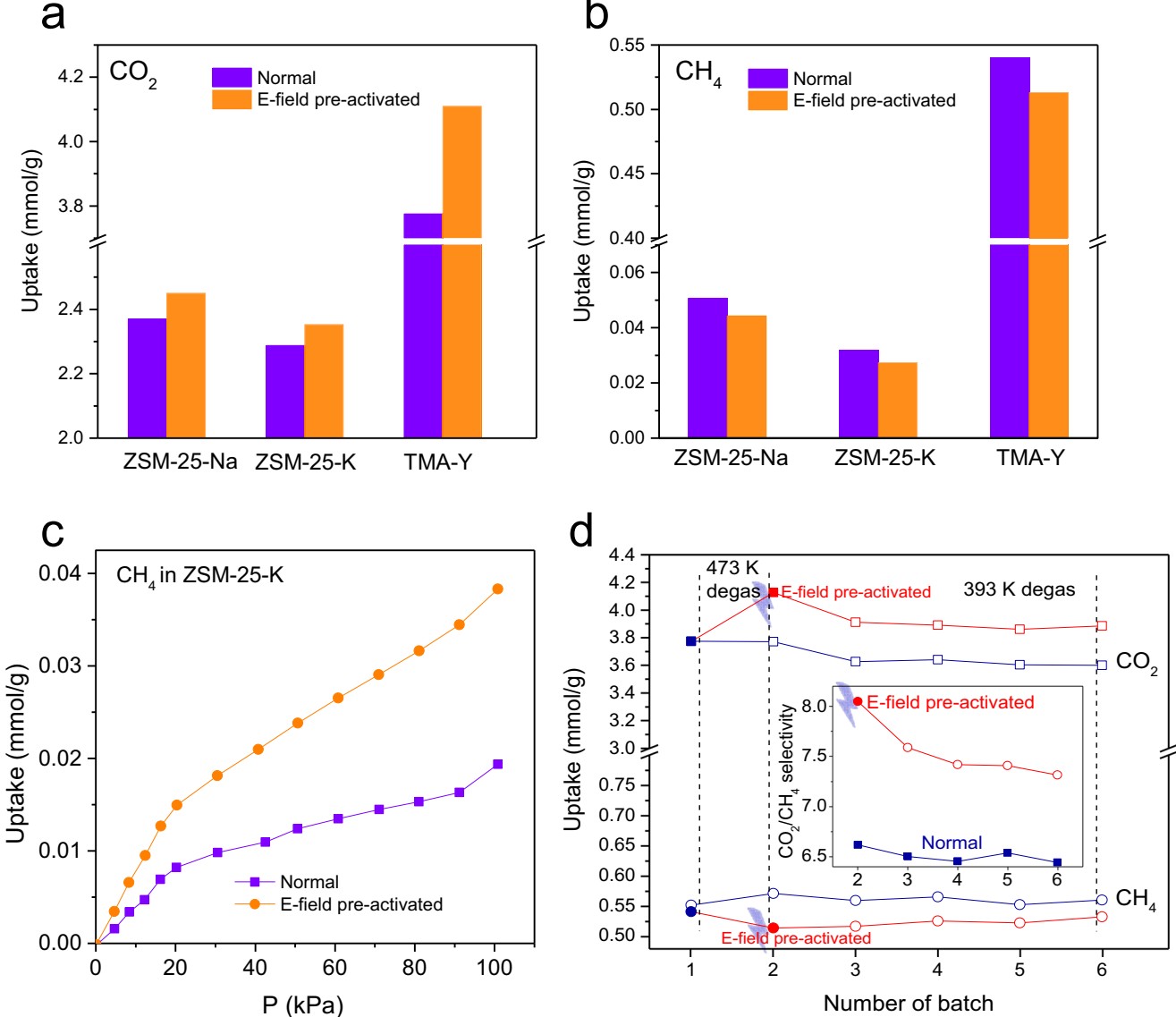

**Fig. 6 | Effect of E-field pre-activation on gas adsorption in zeolites ZSM-25 and TMA-Y.** Adsorption capacity of (**a**) $CO_2$ and (**b**) $CH_4$ in different zeolites (ZSM-25-Na, ZSM-25-K, and TMA-Y) at 101 kPa and 294 K without and with the E-field pre-activation. The complete adsorption isotherms are in Supplementary Figs. 16–20. **c** Adsorption isotherm of $CH_4$ in ZSM-25-K at 252 K without and with the E-field pre-activation. **d** Repeated adsorption batches of $CO_2$ and $CH_4$ adsorptions in TMA-Y. The E-field pre-activation was only conducted before the second batch of adsorption. The temperature represents the degassing temperature between two batches of adsorption trials. Source data are provided as a Source Data file.

adsorption and accelerated its elution process (Fig. 7), which was consistent with the single-component adsorption isotherm. The breakthrough of $CO_2$ did not show remarkable change after the E-field pre-activation of the adsorbent, which resulted from the competitive adsorption between $CH_4$ and $CO_2$, where the adsorption capacity of the weak component $CH_4$ is further depressed in the presence of the strong component $CO_2$. The pre-activation by E-field amplified the suppression of $CH_4$, leading to a sixfold increase of $CO_2/CH_4$ selectivity (calculated by the mass ratio of $CO_2/CH_4$ in the adsorbed gas mixture when the breakthrough $CO_2$ concentration was 5%). The $CO_2$ adsorption capacity of TMA-Y calculated based on the area integral under the breakthrough curve increased from 90.77 to 99.56 mL/g after the E-field pre-activation. These results corresponded with our prediction based on the single-component adsorption performance of TMA-Y and confirmed that the E-field pre-activation approach is effective in improving gas separation selectivity.

## Discussion

We report an E-field pre-activation approach to regulating the gas adsorption capacity of zeolite molecular sieves. Such pre-activation involves applying an external E-field to zeolites during degassing, leading to the relocation of extra-framework cations and the expansion of the zeolite framework. After the E-field pre-activation, the vacant cation sites can be new adsorption sites for $CO_2$ molecules, increasing the $CO_2$ adsorption capacity. The framework expansion and the decrease of cation-guest interaction after the cation relocation resulted in the reduced adsorption energy of gas molecules, decreasing the adsorption capacities of $CH_4$ and $N_2$. Consequently, the adsorption selectivity of $CO_2$ towards $CH_4$ and $N_2$ can be significantly improved by E-field pre-activation in various zeolites, including chabazite, zeolite ZSM-25, and zeolite TMA-Y. Meanwhile, the effect of E-field pre-activation can be maintained for several cycles of adsorption-desorption by degassing the adsorbent at a moderate temperature. After a one-off E-field pre-activation, the

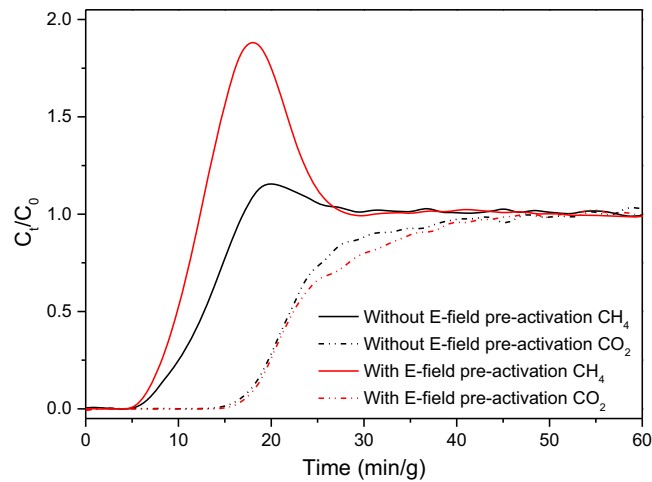

**Fig. 7 | Experimental binary breakthrough curves for a dry gas mixture of $CO_2$/ $CH_4$ (50:50, v/v) on TMA-Y zeolites at 298 K and 101.3 kPa.** $C_t$ and $C_0$ denote the outlet and inlet concentrations, respectively. The red line denotes the TMA-Y sample has been pre-activated by an E-field of 800 V/mm and 4000 Hz, while the black line means the TMA-Y sample did not receive any E-field pre-activation. Source data are provided as a Source Data file.

$CO_2$/$CH_4$ selectivity of TMA-Y remained higher than that without E-field pre-activation in five subsequent adsorption tests. The gas separation selectivity of TMA-Y zeolite for $CO_2$/$CH_4$ in the binary breakthrough experiments showed a sixfold improvement by the E-field pre-activation. Notably, the E-field induced cation relocation can also provide a "gate-opening" effect to trapdoor zeolites, facilitating the admission of guest molecules that were originally blocked by the trapdoor ions. These findings reveal an E-field induced structural transition of zeolite that has never been discovered before. They also demonstrate how to use E-field to adjust the pore accessibility of microporous materials, which provides an approach to improving molecular separation and carbon capture.

## Methods

### Zeolite synthesis
Three zeolites were prepared for this study: Chabazite, ZSM-25, and TMA-Y.

Chabazite with the Si/Al of 2.2 was synthesized from HY zeolite (CBV400) following the reported procedure[3]. It was ion-exchanged by KCl solution twice to obtain fully exchanged potassium chabazite (r2KCHA). Specifically, 5 g of chabazite was refluxed in 200 mL KCl (1 M) at 343 K for 24 h and then filtrated with deionized water 3 times.

ZSM-25-Na zeolite was synthesized following the reported method[36]. ZSM-25-K zeolite was obtained by the ion exchange treatment of ZSM-25-Na. To be specific, 1 g of ZSM-25-Na was dispersed in 2 mL of 1 M KCl solution at room temperature for 2 h, and then filtrated with deionized water 4 times and dried at 353 K overnight. The ion exchange procedure was repeated 6 times to get a fully exchanged ZSM-25-K.

TMA-Y zeolite was prepared from NaY zeolite following the reported method, with ~30% of $Na^+$ exchanged by the organic cation $[C_4H_{12}N]^+$[34].

### Sample degassing
Before the E-field pre-activation and gas adsorption measurements, zeolite powders were tableted into thin plates (0.5 mm in thickness) by applying a pressure of 7000 psi for 3 min, cut into the size of 15 mm (L) × 5 mm (W), and heated stepwise to be thoroughly degassed under

vacuum (<1.2 Pa) overnight. Specifically, the heating rate was 10 K/min, and the sample was kept for 60 min at 323 K, 373 K and 423 K, respectively before reaching the degassing temperature. The degassing temperatures for r2KCHA, ZSM-25, and TMA-Y were 623 K, 573 K, and 473 K, respectively, which were determined by their structural stabilities at high temperatures.

### E-field pre-activation of zeolites
The E-field pre-activation of the zeolite samples was conducted during degassing. The zeolite layer was sandwiched between two porous stainless-steel plates which acted as electrodes and clipped by a foldback clip which was separated from the electrodes by two pieces of glass. They were placed into a two-fold glass cell that can directly connect to the gas-dosing manifold of the adsorption measurement equipment. The glass cell was inserted with two metal hooks to hang the nichrome wires welded to the electrodes. The joint part of the glass cell was sealed with vacuum grease. The specific appearance of the sample and electrodes is shown in Fig. 1a. An external power source (Asterion Programmable AC/DC Power Source, California Instruments, USA) was connected to the metal hooks on the glass cell by wires to apply an E-field of 800 V/mm (400 V between 0.5 mm) to the zeolite layer under high vacuum (0.01 Pa). For the E-field pre-activation, two batches of r2KCHA were exposed to two kinds of E-fields (a 4000 Hz alternating current E-field and a direct current E-field) at 573 K to investigate the influence of the current type. The TMA-Y was pre-activated by an alternating current E-field of 4000 Hz at 473 K. The ZSM-25 was pre-activated by a direct current E-field at 573 K. The procedure of the E-field pre-activation is shown in Fig. 1b. After 30 min E-field pre-activation, the sample was rapidly cooled down under a vacuum by removing the heating jacket. The E-field was removed after the sample was cooled to room temperature.

### Gas adsorption measurements
The adsorption isotherms of $CO_2$, $CH_4$ and $N_2$ were recorded by a Micromeritics ASAP 2010 system. The equilibrium interval for $CO_2$ was 60 s, and that for $CH_4$ and $N_2$ was 30 sec. During gas adsorption analysis, the sample temperature was controlled by a Dewar filled with water for 294 K, ice water for 273 K, and sodium chloride-ice mixture for 252 K. By supplementing the melted ice on time, the fluctuation of the temperature was strictly controlled within 1 K. Each set of adsorption measurements was conducted three times to quantify the error. Firstly, the normal adsorption isotherm of the zeolite sample without an E-field pre-activation was measured. Then the same sample was degassed again after adsorption and then pre-activated by the E-field before the second batch of gas adsorption measurements, which provided the adsorption isotherm with the E-field pre-activation. As a comparison, an in situ E-field of 400 V/mm was applied during the gas adsorptions of r2KCHA.

### Breakthrough gas separation
The dynamic gas breakthrough experiment of TMA-Y for the dry mixture of $CO_2$/$CH_4$ (50:50, v/v) was carried out in a stainless-steel column (3.3 mm inner diameter × 15.8 mm) at 298 K under 1 atm. The gas flow rates were all controlled by mass flow controllers (D07-7B, Beijing Sevenstar Flow Co., Ltd.). TMA-Y samples with or without E-field pre-activation were previously degassed in the specialized tubes and then transferred into the breakthrough stainless steel column in the glove box. For a typical breakthrough experiment, 0.32 g of TMA-Y sample with a height of ~5.3 mm were packed into the column and firstly purged with He flow for 1 h at 298 K. Then the gas mixture of $CO_2$/$CH_4$ with the flow of 5 mL/min was introduced into the column. The $CO_2$ and $CH_4$ concentrations in the outlet gas were detected using a gas analyzer and mass spectrometer (BELMass, BEL JAPAN. INC.) within 60 min.

## Surface area analysis

The specific surface area of r2KCHA without/with the E-field pre-activation was determined by the Micromeritics 3Flex Adsorption Analyzer with $N_2$ at 77 K. After being pre-activated by the 800 V/mm E-field, the r2KCHA was transferred into the sample tube of Micromeritics 3Flex in the $N_2$ atmosphere for surface area analysis. The normal r2KCHA without E-field pre-activation was analyzed together as a comparison. The surface area was calculated by a standard Brunauer–Emmett–Teller (BET) method[37] within the pressure range of $0.05 < P/P_0 < 0.3$.

## Powder X-ray diffraction analysis

Powder X-ray diffraction (PXRD, Bruker D2 phaser) using Cu Kα radiation was used for the identification of possible framework transformations of r2KCHA after the E-field pre-activation. It was conducted in the air at room temperature. The lattice constants of the normal and the E-field pre-activated r2KCHA were obtained by cell refinement method using MDI Jade 6.0 software.

The in situ synchrotron PXRD analysis was conducted at PD beamline, Australian Synchrotron, ANSTO with the E-field application during analysis to identify the instant response of zeolites to the E-field. The X-ray wavelength was calibrated to be 0.7732 Å. The degassed zeolite powders were filled into a 0.7 mm quartz capillary and then wax-sealed. Before analysis, the sealing wax was removed, and the capillary was mounted on the PXRD diffractometer to re-degas for 30 min. During analysis, external E-fields with intensities of 100, 200, and 267 V/mm (with voltages of 120, 240, and 320 V, respectively) were applied to the sample by the two-wire electrodes (1.2 mm apart) parallel to the capillary. The PXRD data were collected over the temperature range of 203 - 493 K under vacuum, with a heating/cooling rate of 5 K/min. For each temperature, the sample was successively exposed to the E-fields with the voltages 0, 120, 240, 320, and 0 V. For each setpoint, the sample was held for 60 s and then scanned for 240 s.

## Ab initio density functional theory calculation

The periodic density functional theory (DFT) calculation was conducted using the Vienna ab initio Simulation Package (VASP) V5.0.4[38]. The initial crystal structure of r2KCHA was a rhombohedral lattice obtained from the Rietveld results of synchrotron PXRD data[9]. One r2KCHA unit cell is composed of three double six-ring prisms or one and a half supercavities. The generalized gradient approximation (GGA) and the projector augmented waves (PAW) approach[39] were employed for the DFT calculation. The geometry optimization was performed with the Perdew-Burke-Ernzerhof (PBE) exchange-correlation functional. A gamma point only k-point mesh was used for a single unit cell. The cut-off energy of the plane wave basis-set was 405 eV, and a Gaussian smearing of 0.01 eV was applied. The atomic positions of r2KCHA were optimized until the forces acting on atoms were below 0.015 eV/Å, as suggested by the previous studies[40]. The DFT-D3 functional (IVDW = 11)[41] was applied to account for the van der Waals interaction. The structure of r2KCHA when the $K^+$ was located at different sites was not directly calculated by relaxing the unit cell, because the incomplete plane-wave basis set with respect to cell volume change will arouse the error of "Pulay stress"[42]. To reduce the Pulay stress, we performed the calculations at different volumes by changing the unit cell volume manually. The energy profile of the relative total energy versus the unit cell volume was plotted to obtain the optimized unit cell volume with the minimum total energy. The energy profile for the cation moving from SIII' to SI was calculated by the nudged elastic band method[43]. The spontaneous polarization was calculated by using the Berry phase method[44]. The core electron density of the atoms was obtained by reconstructing all-electron charge density (the LAECHG tag) in VASP[45], and the difference-electron density was calculated as the difference between the electron density

of the r2KCHA-gas complex and the electron density of r2KCHA augmented by that of an isolated gas molecule[28].

## Data availability

The data generated in this study are provided in the Source Data file. Source data are provided with this paper.

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

## Acknowledgements

This work is sponsored by the Australia Research Council (no. DP190101336, G.K.L., P.A.W., and R.Q.S.). It was undertaken on the Powder X-ray Diffraction beamline at the Australian Synchrotron, part of ANSTO. The simulation was carried out at the Shanxi Supercomputing Center of China, and the calculations were performed on TianHe-2.

## Author contributions

G.K.L. and K.C. conceived the idea. K.C. and Z.Y. conducted the experiments and analyzed the data. K.C. and S.H.M. conducted the computational simulation and analyzed the data. R.S. and Q.G. helped with characterizations. K.C. drafted the manuscript. This project was supervised by G.K.L., P.A.W., and R.Q.S.

## Competing interests

The authors declare no competing interests.
