## [Peer review file · Nature Communications]

REVIEWER COMMENTS

Reviewer #1 (Remarks to the Author):

The authors present a study on electric field stimulated gas adsorption in zeolite frameworks. This study is presented extremely well, and I especially like the literature review about this complicated and rarely investigated topic. The authors show experimentally that electric fields are able to expand chabazite frameworks and regulate and also increase the capacity of adsorption and sorption selectivity. The authors show that a relaxation and reorientation of the cations in the zeolite framework takes place and determine exactly what is happening in the zeolite framework combining PXRD and ab initio methods. Last but not least, the authors demonstrate that this effect is not limited to chabazite, but also occurs in ZSM-5.

The E-field pre-activation seem to be enormously useful for adsorptive gas separation also in the breakthrough curve in the end.

Overall, this paper leaves me here very excited. It is enormously well made. I have only 1 tiny question: In zeolites, the heat of adsorption is a very important factor. If you have a packed bed of zeolites, the heat of adsorption tends to influence the adsorption further. If you place a single crystal on aluminium foil, its heat can dissipate very quickly, making it adsorb more gas and faster. How did you rule out any mistakes in your measurement that can potentially happen due to heat of adsorption related changes?

I am certain this paper will have an insane impact on the whole community. I recommend this paper for publication as it is groundbreaking, innovative, well conducted and the research perfectly executed to the point.

Reviewer #2 (Remarks to the Author):

The paper by Chen et al. describes the unprecedented observation of a modulation of the adsorption characteristics of a porous material by an electric field. For a zeolite with extra framework cations, they observe an increased CO₂ adsorption but a reduction for methane and N₂, enhancing the CO₂/CH₄ selectivity by a substantial amount. The field is not acting during adsorption but at elevated temperatures during degassing. The authors describe the effect as a redistribution of the extra-framework cations, leading also to an increase of the lattice parameters. The latter could be corroborated via periodic DFT calculations, supporting the analysis. Structural transformations triggered by electric fields have been proposed based on theoretical calculations, however, for very high electric fields way beyond the breakthrough voltages of the adsorbed gases (and materials), whereas experimental evidence of electric field stimulated adsorption is rather scarce. Thus, especially the

experimental results in this paper are very interesting for the community and also a wider audience and I recommend its publication. However, in my opinion, the theoretical calculations in this work and correspondingly the analysis of the underlying mechanism have a number of shortcomings. I think this system is particularly interesting since it might be possible to achieve a resolved picture of the actual mechanism (in contrast to e.g. Ref. 13, where in my opinion the mechanistic explanation is not convincing). Therefore I would like to encourage the authors to invest some more effort since this would further strengthen the paper.

Major Point:

The authors describe the process during degassing under the electric field as a reorientation of the cation locations. In a way, this is a phase transformation under the electric field, which should lead to a polarization of the system. If this transformation is retained in the absence of the field, there should be a remnant polarization in the system and it should be considered as a ferroelectric material. A quick survey of the literature reveals that there are a number of ferroelectric MOFs and zeolites reported. Also, a relation to perovskites (or perovskite-like framework materials) should be drawn.

For the current work, I strongly suggest to compute the polarization of the system resulting from cation relocations. Localization of electronic states and a proper Berry-Phase treatment should be straightforward in VASP.

I am also not convinced by the scenario sketched in Fig. S20. How can the barriers of relocation be reduced by a field (and at the same time the final sites are energetically equal)? Why should the cations relocate if this is energetically uphill? This scenario could be proven or disproven by corresponding calculations.

Overall, I think the experimental finding is extremely important but the interpretation is not fully convincing in my view.

Minor points:

- Not all readers are familiar with the zeolite structure. It would be helpful for the understanding to explain in more detail: how many equivalent sites of type SIII', SIII etc. exist per unit cell and how many extra framework cations exist. From the current description, it is not exactly clear if the initial state with all cations in SIII' is structurally disordered or not.

- In Fig. S12 energies of the different cation locations are all relative to their local minimum. I suggest to plot these energies all with respect to the global minimum with all K⁺ in the SIII' site in order to reveal also the energetics of relocations. I would also discuss this aspect in the main manuscript. Currently, I can only extract a 0.9 eV uphill process from SIII' → SIII from Figure S21.

- Why did the authors inconsistently change the level of theory with the dispersion correction? For the energetics of lattice expansion, it was not used, but for the guest molecule adsorption, it was included. In my view, it should always be included.

- For the use of the PAW method, a citation of the seminal work by Peter Blöchl is in my view highly justified.

- When comparing the energetics of systems with different lattice sizes with plane wave methods, it is known that discontinuities can arise due to changes in the representation of wavefunctions and densities on the Fourier grids. To my knowledge in VASP tools are available to correct this. Have these methods been used?

- On p. 3, line 69 "filed" -> "field"

- On p. 12, line 220 (Figure caption) "€" -> (e)

Reviewer #3 (Remarks to the Author):

Please find the review below.

In the work **Regulating Adsorption Performance of Zeolites by Pre-activation in Electric Fields** K. Chen et al studied the influence of electric field on CH₄, CO₂ and N₂ adsorption in a series of zeolites. The authors demonstrated that at certain temperatures, the electric field does the opposite effect on carbon dioxide (increasing) and methane (decreasing) adsorption capacity. An important role in the article is given to the description and explanation of the observed phenomenon. The authors conclude that an ion relocation leading to the framework expansion is the main factor affecting the adsorption process. The argument is based on the combination of different experimental methods (equilibrium and dynamic gas adsorption, PXRD) accompanied by *ab initio* DFT calculations. I believe that the work may be of interest to the broad scientific community and additionally motivate further research on the topic. However, there are some issues that should be resolved before the acceptance of the manuscript.

- The details of BET calculations are lacking, as well as adsorption isotherms of N₂ at 77K are not provided. Also, the resulting surface areas are typical for non-porous materials (in both cases before and after pre-activation with an electric field). It could be a sign that the pre-activation affects only the zeolite surface region or that nitrogen is not a suitable gas for the surface area characterization.
- The ions relocation is discussed mainly assuming that K⁺ is moved to the SI cation site. The DFT calculations in the main text also were done for that case. What is the motivation for the cation site choice? Also, the energy profile on S21 is shown for another path: from SIII' to SIII. What is the reason for choosing different sites for the calculations?
- The heat of adsorption for methane and carbon dioxide adsorption is presented in Fig. S8. The shape of the curves is a bit strange and requires some explanation, while it can be a result of erroneous measurements at low pressures. Also, the authors should provide the tabulated data of the isotherms.
- Please provide the details on selectivity calculations.
- Was the zeolite insulated during the E-field pre-activation? If not, could the direct contact with the electrodes lead to the current in the cell? From the text, it is not clear and the authors discuss only the effect of an electric field.
- Do the CH₄ isotherms at 273K and 294K have a crossover at approx. 80 kPa due to the trapdoor effect? What was the equilibration time during the experiment?
- At least some of the isotherm (and better all of them) should be provided in the tabulated form accompanying with the error analysis. That may help to convince the readers that the observed effects are not within the error bars.

There are some minor problems with the text:

- Line 220: Some undefined symbol appears probably instead of (e)
- Line 134: The authors discuss the heat of adsorption for all three gases, but the reference figures contain only data for the two of them.
- Line 302 in SI: Authors described the shape of the isotherms as a Type-II-like, however IUPAC classification is done for non-deforming adsorbents. It is better to rephrase the sentence.

Reviewer #1:

The authors present a study on electric field stimulated gas adsorption in zeolite frameworks. This study is presented extremely well, and I especially like the literature review about this complicated and rarely investigated topic. The authors show experimentally that electric fields are able to expand chabazite frameworks and regulate and also increase the capacity of adsorption and sorption selectivity. The authors show that a relaxation and reorientation of the cations in the zeolite framework takes place and determine exactly what is happening in the zeolite framework combining PXRD and ab initio methods. Last but not least, the authors demonstrate that this effect is not limited to chabazite, but also occurs in ZSM-5. The E-field pre-activation seem to be enormously useful for adsorptive gas separation also in the breakthrough curve in the end.

Overall, this paper leaves me here very excited. It is enormously well made. I have only 1 tiny question: In zeolites, the heat of adsorption is a very important factor. If you have a packed bed of zeolites, the heat of adsorption tends to influence the adsorption further. If you place a single crystal on aluminum foil, its heat can dissipate very quickly, making it adsorb more gas and faster. How did you rule out any mistakes in your measurement that can potentially happen due to heat of adsorption related changes?

I am certain this paper will have an insane impact on the whole community. I recommend this paper for publication as it is groundbreaking, innovative, well conducted and the research perfectly executed to the point.

Thank you for your comments. Exactly, physical adsorption is an exothermic process, which will release heat and cause a temperature rise. Therefore, during adsorption measurements, the sample was palletized in a 0.5 mm thin slice and clamped by a pair of steel plates (i.e., electrodes) which allows for fast dissipation of heat. Also, the measurement cell was placed in a thermostat Dewar flask to avoid temperature fluctuation, as shown in the setup diagram in Figure 1a. The Dewar flask was filled with water, ice water, or ice and salt to keep a constant temperature during adsorption. Since the sample amount used in the adsorption measurement is tiny, a constant temperature bath is enough to control the temperature. We also timely supplemented the melted ice if the adsorption time is long. According to the temperature monitoring, the temperature fluctuation during adsorption was less than 1 °C.

A more detailed description was added to the second paragraph on Page 4, as well as the second paragraph on Page 20: **The thin slice of the zeolite and the pair of steel plates (i.e., electrodes) allowed for fast dissipation of heat. By supplementing the melted ice on time, the fluctuation of the temperature was strictly controlled within 1 °C.**

Reviewer #2:

The paper by Chen et al. describes the unprecedented observation of a modulation of the adsorption characteristics of a porous material by an electric field. For a zeolite with extra framework cations, they observe an increased CO₂ adsorption but a reduction for methane and N₂, enhancing the CO₂/CH₄ selectivity by a substantial amount. The field is not acting during adsorption but at elevated temperatures during degassing. The authors describe the effect as a redistribution of the extra-framework cations, leading also

to an increase of the lattice parameters. The latter could be corroborated via periodic DFT calculations, supporting the analysis. Structural transformations triggered by electric fields have been proposed based on theoretical calculations, however, for very high electric fields way beyond the breakthrough voltages of the adsorbed gases (and materials), whereas experimental evidence of electric field stimulated adsorption is rather scarce. Thus, especially the experimental results in this paper are very interesting for the community and also a wider audience and I recommend its publication. However, in my opinion, the theoretical calculations in this work and correspondingly the analysis of the underlying mechanism have a number of shortcomings. I think this system is particularly interesting since it might be possible to achieve a resolved picture of the actual mechanism (in contrast to e.g. Ref. 13, where in my opinion the mechanistic explanation is not convincing). Therefore, I would like to encourage the authors to invest some more effort since this would further strengthen the paper.

Major Point:

The authors describe the process during degassing under the electric field as a reorientation of the cation locations. In a way, this is a phase transformation under the electric field, which should lead to a polarization of the system. If this transformation is retained in the absence of the field, there should be a remnant polarization in the system, and it should be considered as a ferroelectric material. A quick survey of the literature reveals that there are a number of ferroelectric MOFs and zeolites reported. Also, a relation to perovskites (or perovskite-like framework materials) should be drawn. For the current work, I strongly suggest computing the polarization of the system resulting from cation relocations. Localization of electronic states and a proper Berry-Phase treatment should be straightforward in VASP.

I am also not convinced by the scenario sketched in Fig. S20. How can the barriers of relocation be reduced by a field (and at the same time the final sites are energetically equal)? Why should the cations relocate if this is energetically uphill? This scenario could be proven or disproven by corresponding calculations. Overall, I think the experimental finding is extremely important, but the interpretation is not fully convincing in my view.

Thanks for the suggestion. We calculated the spontaneous polarization of r2KCHA by following the Berry phase method¹ suggested by the reviewer. The polarization difference resulting from the relocation of one K^+ from SIII' to SI was 1.2, 1.1, and 2.9 C/m² along the x, y, and z axes, respectively. It was 10 times higher than the spontaneous polarization of sodalite at room temperature², suggesting that an E-field pre-activation can lead to a remnant polarization in r2KCHA.

The relevant content has been added to the manuscript at the end of Page 10: **By employing the Berry phase method, the spontaneous polarization of r2KCHA induced by the cation relocation was calculated. The polarization difference resulting from the relocation of one K^+ from SIII' to SI was 1.2, 1.1, and 2.9 C/m² along the x, y, and z axes, respectively. It was 10 times higher than the spontaneous polarization of sodalite at room temperature²⁷, suggesting that an E-field pre-activation can lead to a remnant polarization in r2KCHA.**

The corresponding method was added on Page 22: **The spontaneous polarization was calculated by using the Berry phase method.⁴⁴**

As for the question about the previous Figure S20, we agree with the reviewer that the E-field cannot change the energy barrier. The effect of the E-field is to promote the movement of the cation (like the amplified oscillation of the cation at high temperatures) and facilitate the cation to overcome the energy barrier. In the presence of an external stimulus, such as a high temperature, the cations will be able to overcome the energy barrier and move to a new location that has local minimum energy.³ The E-field serves as a novel stimulus to energize the cation relocation in a way similar to a temperature-induced stimulus. Previous studies have proved that the cation movement of zeolites is closely associated with the E-field gradient inside the pores.⁴ Therefore, with the facilitation of an E-field, the cation can overcome the energy barrier to relocate to a local minimum site as shown in the calculated energy profiles in Figure S21 (newly updated).

To correct the error, we removed Figure S20 and added some explanations in the article on Page 15: Based on the experimental phenomena and computational simulation, we believe that an E-field can **facilitate the cation relocation in zeolites. As demonstrated in previous studies, in the presence of an external stimulus, such as a high temperature, the cations will be able to overcome the energy barrier to relocate to a local minimum.**⁹ The E-field serves as a novel stimulus to energize the cation relocation in a way similar to a **temperature-induced stimulus.** When the cation relocates from its pristine location under an E-field, it can hold at its new location after the E-field was removed, because there will be an energy barrier for the cation to move back, as shown by the calculated energy profile in Supplementary Fig.21.

Minor points:

- Not all readers are familiar with the zeolite structure. It would be helpful for the understanding to explain in more detail: how many equivalent sites of type SIII', SIII etc. exist per unit cell and how many extra framework cations exist. From the current description, it is not exactly clear if the initial state with all cations in SIII' is structurally disordered or not.

Thank you for the suggestion. Since the SIII' site is the energetically favored site for cations, which means the energy for the cations to stay at SIII' is lower than at other sites, all cations in SIII' at the initial state are stable and ordered.⁵⁻⁶ In each unit cell of r2KCHA, there are nine extra-framework K⁺, three double-six ring prisms, and one and a half supercavities⁷, thus including three SI sites, six SII sites, nine SIII sites, and nine SIII' sites.

Detailed information about the structure of chabazite was supplemented on Pages 9-10: **Chabazite is a kind of small-pore zeolite consisting of double-six ring (D6R) prisms linked by the tilted four-membered ring (4MR).**²⁴ The three-dimensional structure constitutes a large supercavity accessed by six eight-membered rings (8MR). Each unit cell included three D6Rs and one and a half supercavities.⁷ There are four general cation positions in dehydrated chabazite²⁵: site SI at the center of the D6R prism, site SII at the triad axis of the D6R prism but displaced towards the supercavity, site SIII in the supercavity above the 4MR, and site SIII' in the 8MR (Figure 4a). **Therefore, there are three SI sites, six SII sites, nine SIII sites, and nine SIII' sites in each unit cell. The cation number is dependent on the Si/Al ratio of zeolite, and in r2KCHA, there is a total of nine extra-framework K⁺ in each unit cell. Since monovalent cations, such as K⁺, energetically prefer site SIII',²⁵⁻²⁶ the stable initial structure of r2KCHA unit cell is composed of the framework and nine K⁺ that are all located at SIII'.**

- In Fig. S12 energies of the different cation locations are all relative to their local minimum. I suggest plotting these energies all with respect to the global minimum with all K^+ in the SIII' site in order to reveal also the energetics of relocations. I would also discuss this aspect in the main manuscript. Currently, I can only extract a 0.9 eV uphill process from SIII' \rightarrow SIII from Figure S21.

Thanks for the suggestion. Figure S12 has been revised accordingly on Page 16 in the Supplementary Information.

Figure S12 Changes of relative energy over unit cell volume of r2KCHA with one K^+ moving from SIII' to (a) SIII, (b) SII and (c) SI.

- Why did the authors inconsistently change the level of theory with the dispersion correction? For the energetics of lattice expansion, it was not used, but for the guest molecule adsorption, it was included. In my view, it should always be included.

Thank you for pointing out this problem. Actually, the dispersion correction was also applied in the framework expansion calculation. The dispersion correction was first used in the gas-zeolite system to correct the influence of the Van der Waals interaction between the gas molecule and the framework. But then we found that if the empty zeolite was not corrected with the dispersion correction, the results will not be comparable to those calculated from the gas-zeolite system. Therefore, the dispersion correction was finally applied to all zeolites with or without gas molecules. In the method section, we did not revise the calculation parameters in time. Now it has been corrected as shown on Page 22: **The DFT-D3 functional (IVDW = 11)⁴¹ was applied to account for the van der Waals interaction.**

- For the use of the PAW method, a citation of the seminal work by Peter Blöchl is in my view highly justified.

Thank you for the suggestion. The work by Peter Blöchl was cited in the last line on Page 21: The generalized gradient approximation (GGA) and the projector augmented waves (PAW) approach³⁹ were employed for the DFT calculation.

Reference 39: Blöchl P. E. Projector augmented-wave method. *Phys. Rev. B* **50**, 17953-17979 (1994).

- When comparing the energetics of systems with different lattice sizes with plane wave methods, it is known that discontinuities can arise due to changes in the representation of wavefunctions and densities on the Fourier grids. To my knowledge in VASP tools are available to correct this. Have these methods been used?

Yes, as the plane-wave basis set is not complete with respect to volume change, there will be an error called “Pulay stress” which tends to decrease cell volume. A sufficiently large energy cutoff can help to avoid the Pulay stress. Otherwise, it can also be reduced by performing calculations at different volumes using the same energy cutoff.⁸ In this study, we applied the latter method to calculate the cell volume when the cations were located at different sites. For each cation position, we manually adjusted the cell volume and relaxed the structure to plot the energy curve to find the cell volume at the minimum energy (Figure and S12). The influence of Pulay stress can be eliminated by this method.

Related explanations were added in the Method section on Page 22: **The structure of r2KCHA when the K⁺ was located at different sites was not directly calculated by relaxing the unit cell, because the incomplete plane-wave basis set with respect to cell volume change will arouse the error of “Pulay stress”.⁴² To reduce the Pulay stress, we performed the calculations at different volumes by changing the unit cell volume manually. The energy profile of the relative total energy versus the unit cell volume was plotted to obtain the optimized unit cell volume with the minimum total energy.**

- On p. 3, line 69 “filed” -> “field”

- On p. 12, line 220 (Figure caption) “€” -> (e)

Thank you for correcting these. The typo was revised.

Page 3: Ideally, the E-field required for regulating the adsorption would be below the breakdown voltage of the respective gases.

Page 11: (e) The simulated PXRD patterns of r2KCHA unit cell with eight K⁺ ions located at SIII' site and one K⁺ ion located at SI, SIII and SIII' site, respectively.

Reviewer #3:

In the work **Regulating Adsorption Performance of Zeolites by Preactivation in Electric Fields** K. Chen et al studied the influence of electric field on CH₄, CO₂ and N₂ adsorption in a series of zeolites. The authors demonstrated that at certain temperatures, the electric field does the opposite effect on carbon dioxide (increasing) and methane (decreasing) adsorption capacity. An important role in the article is given to the description and explanation of the observed phenomenon. The authors conclude that an ion relocation leading to the framework expansion is the main factor affecting the adsorption process. The argument is based on the combination of different experimental methods (equilibrium and dynamic gas adsorption, PXRD) accompanied by *ab initio* DFT calculations. I believe that the work may be of interest to the broad

scientific community and additionally motivate further research on the topic. However, there are some issues that should be resolved before the acceptance of the manuscript.

- The details of BET calculations are lacking, as well as adsorption isotherms of N₂ at 77K are not provided. Also, the resulting surface areas are typical for non-porous materials (in both cases before and after pre-activation with an electric field). It could be a sign that the pre-activation affects only the zeolite surface region or that nitrogen is not a suitable gas for the surface area characterization.

Thank you for the suggestion. The details of BET calculation were supplemented in the first paragraph of Page 21: The normal r2KCHA without E-field pre-activation was analyzed together as a comparison. The surface area was calculated by a standard Brunauer–Emmett–Teller (BET) method³⁷ within the pressure range of $0.05 < P/P_0 < 0.3$.

The adsorption isotherms of N₂ at 77 K were presented in Supplementary Information as Figure S13. The calculation followed the standard Brunauer–Emmett–Teller method⁹, so the specific formula was omitted.

Figure S13 Adsorption isotherms of N₂ in r2KCHA at 77 K without and with the E-field pre-activation.

The N₂ adsorption capacity of r2KCHA at 77 K was as low as non-porous materials. Because the K⁺ cations at the center of 8MR served as trapdoor ions blocked the 8MR allowing for little gas admission, as demonstrated in references.^{3, 10-11} The interaction between N₂ and K⁺ is not strong enough to open the trapdoor. After E-field pre-activation, although the N₂ adsorption capacity was still low, it was doubled compared with that without pre-activation. Therefore, the BET results suggested an influence of the E-field on the trapdoor cation, which has been validated by the DFT calculation afterward.

Nonetheless, the N₂ adsorption capacity of the pre-activated r2KCHA still cannot reach the theoretical value, which should be an N₂ capacity of a microporous material. It implied that the applied E-field can only cause the relocation of a fraction of trapdoor cations, as described on Pages 11-12: According to the maximum

adsorption amount of N₂ in r2KCHA regardless of the trapdoor effect (Supplementary Table 3), the relocated K⁺ induced by the E-field pre-activation was estimated to be approximately 12%.

As for choosing the N₂ to evaluate the influence of the E-field pre-activation, the major reason is N₂ is a weak gas that is sensitive to the location of trapdoor cations. At a low temperature of 77 K, it can sensitively reflect the deviation of the cation. To measure the real surface area excluding the factor of the trapdoor effect, a much higher measurement temperature and a stronger gas for probing the pores would be required. For example, CO₂ will be more suitable as it can strongly interact with the trapdoor cations and successfully diffuse into the cavity. A previous study showed the surface area of r2KCHA measured at 273 K temperature based on the CO₂ isotherms was 584.4 m²/g.¹² However, it was beyond the thesis of this article, so we did not mention it.

The following are the supplementations related to this question in the revised manuscript on Page 11: **Therefore, the N₂ surface area of r2KCHA at 77 K is sensitive to the relocation of trapdoor cations.** After being pre-activated by the E-field, **the N₂ adsorption of r2KCHA at 77 K significantly increased (Supplementary Fig.13) and the BET surface area doubled from 11.73 to 21.67 m²/g,** confirming that some of the door-keeping cations have moved away to admit the originally rejected N₂.

- The ions relocation is discussed mainly assuming that K⁺ is moved to the SI cation site. The DFT calculations in the main text also were done for that case. What is the motivation for the cation site choice? Also, the energy profile on S21 is shown for another path: from SIII' to SIII. What is the reason for choosing different sites for the calculations?

Choosing the SI site in Figure 5 was just to exemplify the situation that the SIII' site was vacated and the subsequent influence on the gas molecule. As shown in Figure 4, when the cation relocated from SIII' to other sites, no matter for SI, SII, or SIII, the structure of r2KCHA exhibited a framework expansion that corresponded to the PXRD analysis. However, the current characterization methods are not powerful enough to determine the specific location of the cation after the E-field pre-activation. Therefore, we chose the SI site, which can cause a moderate expansion of the framework as shown in Figure 4c.

As for the energy profile in Figure S21, it came from our previous studies about the trapdoor effect of r2KCHA. To keep consistency in the paper, we replaced the energy profile of SIII' to SIII with SIII' to SI in Figure S21.

Figure S21 The energy profile for one K^+ in r2KCHA relocating between SIII' and SI calculated by DFT.

The relevant explanation was added on Page 12: To explain how the cation relocation and framework expansion influence gas adsorption, we placed the gas molecules into the supercavity of r2KCHA and calculated the adsorption energies with one K^+ located at SIII' and SI, respectively (other K^+ were always at SIII'), as the K^+ relocation to SI could induce a moderate expansion of the framework (Figure 4c).

- The heat of adsorption for methane and carbon dioxide adsorption is presented in Fig. S8. The shape of the curves is a bit strange and requires some explanation, while it can be a result of erroneous measurements at low pressures. Also, the authors should provide the tabulated data of the isotherms.

Thank you for the suggestion. The tabulated data of all the isotherms were provided in the data source accompanied by the manuscript.

The explanations were added on Page 11 in the Supplementary Information: The heat of adsorption curve of CO_2 was first increased at the low coverage, which might be resulted from the strong interaction of the trapdoor cation and the CO_2 molecule at the beginning of adsorption. After a certain coverage, the increase of adsorbate - adsorbate interaction energy was limited by the adsorption capacity, while the drop of field gradient-quadrupole interaction, dispersion energy, repulsion energy, and polarization energy became dominant, finally leading to a decline in the Q_{st} .³

The heat of adsorption curve of CH_4 in a normal r2KCHA showed a slight increase with increasing coverage. However, after the E-field pre-activation, due to the reduced cation-molecule interaction caused by the relocation of partial K^+ in the pore, the Q_{st} of CH_4 in r2KCHA exhibited a declining trend at low coverages.

- Please provide the details on selectivity calculations.

Thanks for the suggestion. The details of the pure component selectivity calculation were provided on Page 6: The pure component selectivities¹⁹ for component A against component B (A/B) was calculated by Equation (1):

$$\text{Selectivity} = \frac{x_A/y_A}{x_B/y_B} \quad (1)$$

where x is the adsorbed-phase concentration and y is the corresponding gas-phase concentration.

The details of the selectivity calculated from the breakthrough curves were provided on Page 17: The pre-activation by E-field amplified the suppression of CH₄, leading to a 6-fold increase of CO₂/CH₄ selectivity (calculated by the mass ratio of CO₂/CH₄ in the adsorbed gas mixture when the breakthrough CO₂ concentration was 5%).

- Was the zeolite insulated during the E-field pre-activation? If not, could the direct contact with the electrodes lead to the current in the cell? From the text, it is not clear and the authors discuss only the effect of an electric field.

During the E-field pre-activation, the zeolite was directly in contact with the electrodes, whereas the electrodes did not contact each other. Since zeolites are dielectric materials, at the conditions in our experiments, there was no current in the cell. According to the previous study, if the temperature and the E-field intensity are very high, zeolites will become conductive.¹³ However, the E-field intensity used in this study was below the electrical breakdown voltage of the zeolite material. When we increased the temperature to 400 °C and the voltage to 800 V (the E-field intensity was 1600 V/mm), an abrupt increase in the current was observed, showing that the bound charge of the zeolite was polarized by the external E-field.

Here we have added some explanations to the manuscript on Page 4: The electrodes were separated by the insulated zeolite plate so that there was no current in the system and the external power source only provided a static E-field.

- Do the CH₄ isotherms at 273K and 294K have a crossover at approx. 80 kPa due to the trapdoor effect? What was the equilibration time during the experiment?

Yes, because of the trapdoor effect, the uptakes of CH₄ at 273 and 294 K were close³, leading to a crossover of the isotherms. The equilibrium interval for CH₄ and N₂ was 30 sec, and that for CO₂ was 60 sec. The total equilibrium time for each point depended on the rate of adsorption of different gases. We supplemented this information on Page 20 in the article and Page 4 in the Supplementary Information: The equilibrium interval for CO₂ was 60 sec, and that for CH₄ and N₂ was 30 sec. Because of the trapdoor effect, the uptakes of CH₄ at 273 and 294 K were close, leading to a crossover of the isotherms.

- At least some of the isotherm (and better all of them) should be provided in the tabulated form accompanying with the error analysis. That may help to convince the readers that the observed effects are not within the error bars.

Thanks for the valuable suggestion. The tabulated data of all isotherms in the manuscript was provided in the data source that was attached as a separate file. Each batch of adsorption measurement was tested at least three times and the errors were provided. The error analysis was also shown in Figure 2d and the explanations were supplemented on Pages 5-6: The E-field effect on a specific gas is consistent at different temperatures with small errors. As shown in Figure 2d the error bars were much smaller than the uptake difference caused by the E-field pre-activation, excluding the possibility that the effect of the E-field was raised from analysis error.

There are some minor problems with the text:

- Line 220: Some undefined symbol appears probably instead of (e)

Thanks for pointing it out. It has been corrected as: (e) The simulated PXRD patterns of r2KCHA unit cell with eight K^+ ions located at SIII' site and one K^+ ion located at SI, SIII and SIII' site, respectively.

- Line 134: The authors discuss the heat of adsorption for all three gases, but the reference figures contain only data for the two of them.

The reference Figure S8 only contained the heat of adsorption curves of CO_2 and CH_4 . Because at the temperature range of 252 to 294 K, the N_2 adsorption in r2KCHA was strongly influenced by the trapdoor effect. The uptake was increased with the increasing temperature, disobeying the pattern of common physical adsorption. The experimentally measured N_2 uptake at these temperatures cannot reflect the real adsorption capacity due to the blocked pores. Therefore, the Clapeyron equation is not valid in the tested temperature region.

However, in Figure S7 and Table S3, the heat of adsorption of N_2 was also calculated by fitting the adsorption isotherms to the LJM-Toth model, an adsorption model that can describe trapdoor adsorption.³ As shown by the Q value in Table S3, the heat of adsorption N_2 was decreased after the E-field pre-activation, which was similar to the situation of CO_2 and CH_4 .

The explanations were added to the Supplementary Information on Page 10: At the temperature range of 252 to 294 K, the N_2 adsorption in r2KCHA was strongly influenced by the trapdoor effect. The uptake was increased with the increasing temperature, disobeying the pattern of common physical adsorption. The experimentally measured N_2 uptake at these temperatures cannot reflect the real adsorption capacity due to the blocked pores. Therefore, its heat of adsorption cannot be calculated through the Clapeyron equation. The heat of adsorption of N_2 can be obtained from the fitted LJM-Toth model, as shown by the parameter Q in Table S3.

- Line 302 in SI: Authors described the shape of the isotherms as a Type-II like, however IUPAC classification is done for non-deforming adsorbents. It is better to rephrase the sentence.

Thank you for the suggestion. The sentence was rephrased on Page 20 in the Supplementary Information: The Na-ZSM-25 exhibited a rapid increase of CO₂ uptake at low pressures and a flat upward trend at high pressures, which was attributed to the framework expansion while adsorbing CO₂.⁴

References:

- (1) King-Smith, R. D.; Vanderbilt, D. Theory of Polarization of Crystalline Solids. *Phys. Rev. B Condens. Matter.* **1993**, *47* (3), 1651.
- (2) Maeda, Y.; Wakamatsu, T.; Konishi, A.; Moriwake, H.; Moriyoshi, C.; Kuroiwa, Y.; Tanabe, K.; Terasaki, I.; Taniguchi, H. Improper Ferroelectricity in Stuffed Aluminate Sodalites for Pyroelectric Energy Harvesting. *Phys. Rev. Appl.* **2017**, *7* (3).
- (3) Li, G. K.; Shang, J.; Gu, Q.; Awati, R. V.; Jensen, N.; Grant, A.; Zhang, X.; Sholl, D. S.; Liu, J. Z.; Webley, P. A.; May, E. F. Temperature-Regulated Guest Admission and Release in Microporous Materials. *Nat. Commun.* **2017**, *8*, 15777.
- (4) Jordan, E.; Bell, R. G.; Wilmer, D.; Koller, H. Anion-Promoted Cation Motion and Conduction in Zeolites. *J Am. Chem. Soc.* **2006**, *128* (2), 558.
- (5) Smith, L. J.; Eckert, H.; Cheetham, A. K. Potassium Cation Effects on Site Preferences in the Mixed Cation Zeolite Li,Na-Chabazite. *Chem. Mater.* **2001**, *13* (2), 385.
- (6) Saxton, C. G.; Kruth, A.; Castro, M.; Wright, P. A.; Howe, R. F. Xenon Adsorption in Synthetic Chabazite Zeolites. *Microporous Mesoporous Mater.* **2010**, *129* (1-2), 68.
- (7) Shang, J.; Li, G.; Webley, P. A.; Liu, J. Z. A Density Functional Theory Study for the Adsorption of Various Gases on a Caesium-Exchanged Trapdoor Chabazite. *Comput. Mater. Sci.* **2016**, *122*, 307.
- (8) Vanpoucke, D. E. P.; Lejaeghere, K.; Van Speybroeck, V.; Waroquier, M.; Ghysels, A. Mechanical Properties from Periodic Plane Wave Quantum Mechanical Codes: The Challenge of the Flexible Nanoporous Mil-47(V) Framework. *J. Phys. Chem. C* **2015**, *119* (41), 23752.
- (9) Fagerlund, G. Determination of Specific Surface by the BET Method. *Matériaux et Construction* **1973**, *6*, 239.
- (10) Shang, J.; Li, G.; Singh, R.; Gu, Q.; Nairn, K. M.; Bastow, T. J.; Medhekar, N.; Doherty, C. M.; Hill, A. J.; Liu, J. Z.; Webley, P. A. Discriminative Separation of Gases by a "Molecular Trapdoor" Mechanism in Chabazite Zeolites. *J Am. Chem. Soc.* **2012**, *134* (46), 19246.
- (11) Shang, J.; Li, G.; Gu, Q.; Singh, R.; Xiao, P.; Liu, J. Z.; Webley, P. A. Temperature Controlled Invertible Selectivity for Adsorption of N₂ and CH₄ by Molecular Trapdoor Chabazites. *Chem. Commun. (Camb)* **2014**, *50* (35), 4544.
- (12) Ridha, F. N.; Yang, Y.; Webley, P. A. Adsorption Characteristics of a Fully Exchanged Potassium Chabazite Zeolite Prepared from Decomposition of Zeolite Y. *Microporous Mesoporous Mater.* **2009**, *117* (1-2), 497.
- (13) Bordeneuve, H.; Wales, D. J.; Physick, A. J. W.; Doan, H. V.; Ting, V. P.; Bowen, C. R. Understanding the AC Conductivity and Permittivity of Trapdoor Chabazites for Future Development of Next-Generation Gas Sensors. *Microporous Mesoporous Mater.* **2018**, *260*, 208.

VIEWERS' COMMENTS

Reviewer #1 (Remarks to the Author):

I remain excited about this study. It is a great piece of work. My comments are answered fully. I have checked on the other review and the answers there as well. The referee had many important points and for me, the authors answered and worked on all the concerns raised.

I suggest acceptance of the manuscript.

Reviewer #2 (Remarks to the Author):

The authors have properly addressed all my points of criticism and in my opinion, also the points raised by the other reviewers have been considered. The revised manuscript is substantially improved with respect to the original version and I recommend its publication of this interesting work as is.

Reviewer #3 (Remarks to the Author):

I have no further comments.